# ELMO1 Deficiency Reduces Neutrophil Chemotaxis in Murine Peritonitis

**DOI:** 10.3390/ijms24098103

**Published:** 2023-04-30

**Authors:** Shuxiang Yu, Xiaoke Geng, Huibing Liu, Yunyun Zhang, Xiumei Cao, Baojie Li, Jianshe Yan

**Affiliations:** 1School of Medicine, Shanghai University, Shanghai 200444, China; 2School of Life Sciences, Shanghai University, Shanghai 200444, China; 3School of Environmental and Chemical Engineering, Shanghai University, Shanghai 200444, China; 4State Key Laboratory Cell Differentiation and Regulation, Henan International Joint Laboratory of Pulmonary Fibrosis, Henan Center for Outstanding Overseas Scientists of Pulmonary Fibrosis, College of Life Science, Henan Normal University, Xinxiang 453007, China; 5Shanghai Institute of Immunology, Department of Immunology and Microbiology, Shanghai Jiao Tong University School of Medicine, Shanghai 200025, China; 6Bio-X Institutes, Key Laboratory for the Genetics of Developmental and Neuropsychiatric Disorders, Ministry of Education, Shanghai Jiao Tong University, Shanghai 200240, China

**Keywords:** ELMO1, neutrophil, chemotaxis, peritonitis, inflammation

## Abstract

Peritoneal inflammation remains a major cause of treatment failure in patients with kidney failure who receive peritoneal dialysis. Peritoneal inflammation is characterized by an increase in neutrophil infiltration. However, the molecular mechanisms that control neutrophil recruitment in peritonitis are not fully understood. ELMO and DOCK proteins form complexes which function as guanine nucleotide exchange factors to activate the small GTPase Rac to regulate F-actin dynamics during chemotaxis. In the current study, we found that deletion of the Elmo1 gene causes defects in chemotaxis and the adhesion of neutrophils. ELMO1 plays a role in the fMLP-induced activation of Rac1 in parallel with the PI3K and mTORC2 signaling pathways. Importantly, we also reveal that peritoneal inflammation is alleviated in Elmo1 knockout mice in the mouse model of thioglycollate-induced peritonitis. Our results suggest that ELMO1 functions as an evolutionarily conserved regulator for the activation of Rac to control the chemotaxis of neutrophils both in vitro and in vivo. Our results suggest that the targeted inhibition of ELMO1 may pave the way for the design of novel anti-inflammatory therapies for peritonitis.

## 1. Introduction

Peritoneal dialysis (PD) is an effective alternative therapy for renal failure, but infection and infection-associated acute inflammation can lead to treatment failure, which is a common complication of PD. The early stages of acute peritonitis are characterized by massive neutrophil infiltration. Neutrophils are the dominant white blood cells in the human body and play a key role in the first-line immune response. Dysregulation of the number or function of neutrophils leads to neutrophil inflammatory diseases such as coronary artery disease [1], rheumatoid arthritis [2], and sepsis [3]. Chemotaxis, the directional movement of cells toward chemokines and other chemoattractants, plays a critical role in diverse physiological processes, such as embryonic development, the targeting of metastatic cancer cells to specific tissues, and the mobilization of immune cells to fight invading microorganisms. Neutrophil chemotaxis to sites of infection and tissue injury is a critical process during immune responses that leads to acute and chronic inflammation. The migration of neutrophils to the site of infection is a four-step process, including neutrophil capture, rolling, stable adhesion, and transendothelial migration [4].

Neutrophil chemotaxis is induced by chemoattractant stimuli, including lipids, N-formylated peptides, complement anaphylatoxins, and chemokines [5,6]. During the inflammatory response, the binding of chemokines to G-protein-coupled receptors (GPCRs) on the surface of immune cells induces the dissociation of the heterotrimeric G protein to two free subunits, Gα and Gβγ [7,8]. Gβγ activates downstream effectors, including Rho family proteins, PI3K/mTORC2, phospholipase C, MAPK, JAK/Stat, and NF-κB, which ultimately leads to cell polarization and migration [9,10,11]. Studies have shown that PI3K promotes the polarization phenotype of neutrophils to promote efficient chemotaxis [12,13,14], which depends on the site activation and phosphorylation of the downstream effector AKT threonine-308 [15,16]. However, another study showed that PI3K indeed accelerates neutrophil chemotaxis, but is not required [17]. mTORC2 is a key regulator of the actin cytoskeleton [18]. By acting downstream of Ras, it regulates the phosphorylation of AKT serine-473 and participates in the chemotaxis of neutrophils [19]. Together, these signaling components lead to the activation of the small GTPase Rac to regulate the reorganization of the actin cytoskeleton that generates directional force for cell chemotaxis at the leading edge. At the trailing end of the polarized neutrophils, activated RhoA stimulates the assembly of actomyosin contractile complexes to detach the cells from the substratum as a result of RhoA/ROCK-dependent actomyosin contractile forces [20].

The engulfment and cell motility protein (ELMO) is an evolutionarily conserved adapter protein with three isoforms of ELMO proteins, ELMO 1–3. The ELMO1–DOCK–Rac1 axis has been found to regulate the actin cytoskeleton during osteoclast precursor cell migration [21], the engulfment of apoptotic cells [22], and neurite outgrowth [23]. In previous studies, we showed that ELMO family proteins bind to heterotrimeric G proteins, such as Gα and Gβγ subunits, leading to the activation of Rac GTPase to regulate F-actin dynamics during *D. discoideum* [24] and cancer cell migration [25,26]. Recently, a number of studies have suggested that ELMO1 is involved in inflammatory diseases [21,26,27,28,29,30]. However, the role and mechanism of ELMO1 in regulating the chemotaxis of neutrophils in acute peritonitis remains unclear. By using the adhesion sites at the leading edge as cortical anchors for F-actin polymerization, cells reorganize the cytoskeleton to enhance membrane protrusion during chemotaxis [31]. In this study, we set out to investigate the function of ELMO1 in the adhesion and chemotaxis of neutrophils. Our data show that ELMO1 is required for the migration and adhesion of neutrophils. More importantly, we found that peritoneal inflammation is alleviated in Elmo1 knockout mice in the mouse model of thioglycollate (TG)-induced peritonitis. Together, our results suggest that ELMO1 functions as a conserved regulator to control the chemotaxis of neutrophils both in vitro and in vivo.

## 2. Results

### 2.1. ELMO1 Plays a Role in Acute Peritonitis by Regulating the Infiltration of Neutrophils

The early stage of acute peritonitis is characterized by neutrophil infiltration at the site of infection. To study the possible role of ELMO1 in the inflammatory response, we successfully constructed an acute peritonitis model using 4% TG in Elmo1^−/−^ and Elmo1^+/+^ adult mice. H&E-stained sections revealed that the inflammatory response was reduced in the liver and peritoneum in KO mice compared to WT mice (Figure 1A). The peritoneal lavage fluid after TG induction was collected and analyzed by flow cytometry and ELISA. The results showed that the peritoneal lavage fluid of Elmo1^−/−^ mice had lower cellular accumulation and levels of inflammatory factors (TNF-a and IL-6) than wild-type mice (Figure 1B–D). Immunofluorescent staining of immune cells revealed reduced numbers of Gr-1^+^ neutrophils in the liver and peritoneum in the Elmo1^−/−^ mice (Figure 1E,F). These results suggest that a reduction in the inflammatory response may be attributable to the decreased enrichment of neutrophils in Elmo1^−/−^ mice.

### 2.2. Elmo1 Deficiency Reduces the Chemotaxis and Adhesion of Neutrophils

Studies have shown that under the stimulation of fMLP, Rho, Rac, and Cdc42 are temporarily activated in neutrophils; this is especially true for Rho and Rac, which control skeleton reorganization, reversible adhesion, and directional migration [32,33,34]. To check whether ELMO1 is involved in the chemotaxis of neutrophils, we collected cells in the TG-induced peritoneal lavage fluid and performed flow cytometric analysis. The results showed that Elmo1^−/−^ mice had a lower percentage of neutrophils in the peritoneal lavage fluid than wild-type mice (Figure 2A,B). We next examined the chemotaxis of isolated primary neutrophils with high purity (Appendix A) stimulated with fMLP by transwell assay. Flow cytometry analysis showed that the migration ability of neutrophils increased in a concentration-dependent manner under the stimulation of fMLP, reaching the maximum level at 1 μM. The migration of neutrophils in the Elmo1^−/−^ group was significantly lower than that of the wild-type group (Figure 2C). HL-60, a human leukemia cell line, is a well-established model for studying neutrophil functions, including migration and phagocytosis. To further verify the effect of Elmo1 gene ablation on cell migration, we successfully knocked down Elmo1 in HL-60 cells (Appendix A). We added 1 μM of fMLP to stimulate the HL-60 cells in the transwell experiment and found that the migration ratio of the knockdown group was significantly lower than that of the control group as well (Figure 2D).

Cell adhesion is closely related to the process of cell migration [35]. We next examined the effect of ELMO1 on the adhesion of neutrophils. By measuring the OD value of the absorbance of the adherent cells at 570 nm, we found that with the increase in the fMLP concentration, the adhesion ability of neutrophils went up, reaching the maximum at 10 μM. The adhesion ability of the neutrophils of Elmo1^−/−^ mice was reduced compared with that of wild-type mice (Figure 2E). To further verify the effect of Elmo1 on neutrophil adhesion, we also performed the same assay in HL-60 cells. We found that Elmo1 KD cells showed reduced adhesion (Figure 2F). In addition, we tested the chemotaxis and adhesion of neutrophils and HL-60 under the stimulation of C5a. We found that the chemotaxis and adhesion of Elmo1-deficient cells were reduced compared to those of wild-type cells (Appendix A), indicating that the defect in chemotaxis and adhesion is not limited to a specific agent. Overall, the above results indicate that ELMO1 positively regulates the chemotaxis and adhesion of neutrophils in vivo and in vitro.

### 2.3. Deficiency of Elmo1 Down-Regulates fMLP-Induced Actin Polymerization

The local polymerization of actin causes cell polarization to produce traction and thrust in the process of neutrophil migration. Phalloidin binds the actin between polymerized subunits with high specificity. We used Alexa Fluor 633-phalloidin to stain F-actin, which was followed by flow cytometry to measure the mean fluorescence intensity. In addition, we used confocal microscopy to detect the F-actin cell end-clustering in neutrophils. The results showed that F-actin was polymerized rapidly and transiently under fMLP stimulation. However, the F-actin polymerization in neutrophils from Elmo1^−/−^ mice was reduced compared to that in neutrophils from wild-type mice (Figure 3A–C). Similarly, we found that Elmo1 KD cells showed reduced F-actin polymerization as well (Figure 3D–F). To identify the neutrophils in the sample, we stained the samples with an antibody against Ly6G (Appendix A). In conclusion, our data suggest that a deficiency of Elmo1 reduces the F-actin polymerization in neutrophils.

### 2.4. ELMO1 Regulates fMLP-Triggered Rac Activation Independent of PI3K and mTORC2 Signaling Pathways

Next, we checked whether Elmo1 plays a role in the migration of neutrophils and actin polymerization through Rac1 signaling. We found that compared to wild-type mice, the level of activated Rac1 (Rac1-GTP) in Elmo1^−/−^ neutrophils was remarkably reduced when stimulated by 10 μM of fMLP, indicating that ELMO1 promotes the activation of the Rac protein (Figure 4A,B). It was found that the binding of chemokines to GPCRs activates the downstream signaling pathways of PI3K and mTORC2, each of which transduces signals to the actin cytoskeleton [36]. We therefore examined whether ELMO1 is involved in modulating the PI3K and mTORC2 classic signaling pathways under fMLP stimulation. We found that phosphorylation at AKT-T308 and AKT-S473, which were activated by PI3K and mTORC2, respectively, showed no significant difference between Elmo1^−/−^ and wild-type neutrophils (Figure 4C–E), which was further confirmed in HL-60 cells (Figure 4F–H). We therefore speculated that ELMO1 regulates the Rac activation of neutrophils independent of the PI3K and mTORC2 signaling pathways.

## 3. Discussion

The results presented here reveal the biological functions of ELMO1 in the chemotaxis of neutrophils in murine peritonitis. We found that inflammation was alleviated in Elmo1 knockout mice in the mouse model of TG-induced peritonitis. We showed that ELMO1-deficient neutrophils exhibit defective capabilities of adhesion and migration. Furthermore, we discovered that ELMO1 functions through Rac1 independently of the canonical PI3K and mTORC2 signaling pathways. Taken together, our data suggest that ELMO1 acts as a regulator of neutrophil chemotaxis during acute inflammation.

Previous studies have shown that GPCRs regulate the ELMO/DOCK complex via heterotrimeric G proteins to activate the small GTPase Rac, which promotes actin polymerization during the chemotaxis of *D. discoideum* [24] and human cancer cells [25,26]. Recently, we found that Elmo1 deficiency suppresses the adhesion and migration of osteoclast precursors, which are critical processes for the information of mature osteoclasts, and results in alleviated bone erosion in Elmo1 knockout mice in a rheumatoid arthritis mouse model [21]. In line with our findings, Arandjelovic reported that Elmo1 knockout neutrophils show significantly reduced migration toward chemoattractants of LTB4 and CXCL1 [27]. In our current study, we showed that the deletion of Elmo1 causes defective chemotaxis in neutrophils and human neutrophil-like HL-60 cells stimulated with fMLP, indicating that the mechanism underlying ELMO1 signaling in neutrophil chemotaxis shares a similarity with other organisms or cells. Therefore, our data support the notion that the ELMO/DOCK complex functions as an evolutionarily conserved guanine nucleotide exchange factor to regulate Rac activation in cell chemotaxis.

Neutrophils, the most abundant white blood cells in human blood circulation, play a critical role in the innate immune response that forms the first line of defense against invading pathogens. However, excessive recruitment of these cells results in tissue damage and inflammatory diseases such as arthritis [27], colitis [37], and sepsis [38]. The results reported herein show that ELMO1 plays a role in the recruitment of neutrophils in a mouse model of TG-induced peritonitis. Our current studies cannot exclude the possible involvement of additional innate immune cells such as macrophages, which may also play a role in the inflammation of peritonitis. However, our results are complemented by a recent report demonstrating that neutrophils are the predominant cells in the model of TG-induced peritonitis compared to macrophages [39]. These data suggest that ELMO1 is a key regulator of neutrophil chemotaxis and may be critical to the early inflammatory response in murine peritonitis. It is well established that neutrophils are the first responders, playing a key role in the immune response to acute infection. However, recent studies have suggested that neutrophils may also be involved in chronic inflammation by exhibiting memory-like inflammatory responses after exposure to bacterial components such as LPS [40], LTA [41], or small extracellular vesicles from the gut [42]. It would be interesting to investigate whether ELMO1 functions as a priming neutrophil function in future studies.

Recent studies have revealed that adenovirus therapy targeting ELMO1 delays the inflammatory response in an arthritis animal model [30], and that the c-terminal helical inhibitory peptide of ELMO1 decreases osteoclast-mediated bone resorption [28]. The studies here may have important implications for targeting ELMO1 to inhibit the infiltration of neutrophils into the peritoneum, which may help patients with kidney failure. Further studies would shed light on the functions and mechanisms of ELMO1 in regulating inflammation, which may hopefully pave the way for the design of novel anti-inflammatory therapies.

## 4. Materials and Methods

### 4.1. Mice

Animal experiments were carried out in accordance with the relevant ethical regulations for animal testing and research, and the protocols were reviewed and approved by the Ethics Committee of Shanghai University (the approval number: No. ECSHU2021-045). Elmo1 knockout mice were purchased from the European Mouse Mutant Archive. Mice were maintained in specific pathogen-free conditions in individual ventilated cages and were housed at 22 ± 2 °C under a 12–12 h light/dark cycle with free access to food and water.

### 4.2. Primary Neutrophil Isolation and Identification

The tibia and femur of mice were washed in PBS containing 100 U/mL of penicillin and 100 μg/mL of streptomycin. Bone marrow fluids were flushed from the bone marrow cavity with HBSS containing 2% bovine serum albumin. Bone marrow cells were resuspended in red blood cell lysis buffer, and then the cell suspension was added to the Percoll^TM^ (170891, GE Healthcare, Chicago, IL, USA) buffer with gradient centrifugation at 1500× *g* for 30 min. The middle cell layer was collected and washed with a wash buffer. The clarified cells were resuspended in an RPMI medium (SH30809, HyClone, Logan, UT, USA) containing 10% fetal bovine serum (FBS) (10099, Gibco, New York, NY, USA) and 1% penicillin and streptomycin.

### 4.3. Cell Culture, Infection, and Differentiation

The cultivation and differentiation of HL-60 cells (purchased from ATCC) were performed as described previously [43]. shRNA sequences targeting human Elmo1 (GenBank, NM_014800) and a negative scrambled sequence were constructed by GeneChem (Shanghai, China). The Elmo1 shRNA sequences were GATAGTTCAAACTTCTATA and CCATGTACACGCGAGATTATA, and the negative scrambled shRNA sequence was CCTAAGGTTAAGTCGCCCTCG. Silencing Elmo1 was achieved by inserting shRNAs into the pLVX vector. Recombinant lentiviruses were produced by co-transfecting 293T cells. Lentivirus particles were used to infect cells with the added HitransG (REVG005, GeneChem, Shanghai, China). After co-culturing for 24 h, cells were sorted with green fluorescence by flow cytometry (Beckman Coulter, Krefeld, Germany). The differentiated HL-60 (dHL-60) cells were used for experiments on day 6 after a 1.3% dimethyl sulfoxide (DMSO) (D8418, Sigma-Aldrich, St. Louis, MO, USA) treatment. Differentiation was verified using CD11b (550019, BD Pharmingen™, Franklin Lakes, NJ, USA) and CD71 (555537, BD Pharmingen™, Franklin Lakes, NJ, USA) antibodies for flow cytometry detection and immunofluorescence.

### 4.4. H&E, Immunofluorescence, and Phalloidin Staining Assay

Primary neutrophils and HL-60 cells were cultured in fibrinogen-coated glass-bottom dishes (801001, NEST, Beijing, China) at a density of 2 × 10^5^ cells per dish. After stimulation, the cells were fixed with 4% paraformaldehyde (PFA) for 10 min and incubated with Alexa Fluor 633-phalloidin (A22284, Life Technologies, Carlsbad, CA, USA) for 30 min, which was followed by 4, 6-diamidino-2-phenylindole (DAPI) (D3571, Life Technologies, USA) staining for 2 min. The cells were imaged using a Zeiss 710 LSM confocal microscope (Zeiss, Oberkochen, Germany). Neutrophils and HL-60 cells were collected and resuspended at 1 × 10^7^/mL. Then, 20 μL of cells were stained with phalloidin after stimulation for the indicated times. The detection of MFI (mean fluorescence intensity) was performed via flow cytometry.

The liver and peritoneum were fixed in 4% PFA overnight at 4 °C. Samples were embedded in paraffin. Six-micrometer-thick sections were cut (Leica Microsystems, Wetzlar, Germany) and stained with a hematoxylin and eosin staining solution, and then photographed under a light microscope (Olympus Microsystems, Tokyo, Japan). Immunofluorescence staining was performed on sections of Gr-1 according to standard protocols [44]. In brief, sections were deparaffinized, rehydrated, and permeabilized with 0.1% Triton X-100 for 30 min. For immunostaining, sections were blocked with 10% goat serum for 60 min, incubated with the anti-Gr-1 antibody (87048S, Cell Signaling Technology, Danvers, MA, USA) overnight at 4 °C, and incubated with a secondary antibody for 1 h at 37 °C; then, they were mounted on ProLong Gold DAPI (P36934, Life Technologies, USA). Images were taken and analyzed using the Olympus DP72 microscope (Olympus Microsystems, Japan).

### 4.5. Generation of the Peritonitis Mouse Model

The peritonitis mouse model was generated with 4% TG (70157, Sigma-Aldrich, USA) according to the protocol as described in [45]. In brief, 2 mL of TG was injected into the abdominal cavity. Organs, serum, and peritoneal lavage fluid of mice were collected within 6 h. The same volume of peritoneal lavage solution was centrifuged at 500 g, and the total cell numbers were counted via the FSC (forward scatter) and SSC (side scatter) of flow cytometry. Cells were incubated with anti-CD11b (17-0118, eBioscience, San Diego, CA, USA) and anti-Gr-1 (11-5931, eBioscience, USA) antibodies to label neutrophils. The percentage of neutrophils for the same total cells were recorded by flow cytometry.

### 4.6. Enzyme-Linked Immunosorbent Assay (ELISA)

The contents of tumor necrosis factor-alpha (TNF-α) and interleukin-6 (IL-6) in the peritoneal lavage fluid were measured by TNF-α (88-7324) and IL-6 (88-7064) ELISA kits (eBioscience Co., USA). First, 96-well plates were coated with the capture antibody and incubated overnight at 4 °C. Then, the standard samples and peritoneal lavage fluid were added and incubated for 2 h at room temperature. The corresponding detection antibody, avidin-horseradish peroxidase, and tetramethylbenzidine (TMB) solution were added. After stopping the reaction, optical absorbance was read at 450 nm and analyzed. The concentration of protein was calculated according to the standard curve.

### 4.7. Adhesion Assay

Primary neutrophils and HL-60 cells adhesion assay were conducted in a 96-well plate. Cells suspended in a 100 μL cell culture medium were seeded in triplicate at a density of 2 × 10^6^ cells per well and incubated with fMLP for 30 min at 37 °C. Then, cells were washed with PBS and fixed with 4% PFA for 10 min after the non-adherent cells were aspirated out. The fixed cells were stained with crystal violet for 10 min and dissolved by sodium dodecyl sulfate (SDS) to measure the absorbance at 570 nm using a microplate spectrophotometer.

### 4.8. Transwell Migration Assay

Primary neutrophil and HL-60 cells migration assays were performed in 24-well cell culture insert paired plates on transwell filters with a pore size of three microns (353096, Becton, Dickinson and Company, New York, NY, USA). First, 5 × 10^5^ cells were suspended in a 200 μL medium and added to the upper chamber of the insert pre-coated with 2.5 μg/mL of fibronectin (F0895, Sigma-Aldrich, Burlington, MA, USA). Then, a 600 μL medium containing 100 μM of fMLP (47729, Sigma-Aldrich, USA) was added to the lower chamber for cell migration. Cells in the lower chamber were collected after 2 h of incubation and counted by flow cytometry. Mobility was defined as the ratio of (experimental group—control group)/control group.

### 4.9. Rac Activity Assay

Rac1 activity assays were generated according to the Rac1 activation assay kit (BK035, Cytoskeleton, Denver, CO, USA). In brief, neutrophils were stimulated with 10 μM of fMLP for the indicated times, and then were disrupted with an ice-cold lysis buffer containing a protease inhibitor cocktail. After repeated freezing and thawing, some of the lysates were taken, stored, and used to detect the total Rac1. The remaining cell lysates were incubated with PAK-PBD beads to detect active Rac1. Samples were analyzed by SDS-PAGE and immunoblotting using the anti-Rac1 antibody.

### 4.10. Western Blot Analysis

Proteins extracted from cells were separated via SDS-PAGE and transferred to polyvinylidene difluoride (PVDF) membranes. The PVDF membrane was blocked with 5% skim milk and probed with antibodies against p-AKT-T308, p-AKT-S473, AKT (2965S, 4060S, and 9272S; Cell Signaling Technology, USA), ELMO1 (ab174298, Abcam, Cambridge, UK), and GAPDH (C1219, Tianjin Sungene Biotech, Tianjin, China) for the indicated proteins.

### 4.11. Statistical Analysis

Statistical significance was evaluated with GraphPad Prism 6 using Student’s 2-tailed, unpaired *t*-test analysis of variance and a one-way ANOVA was used for comparisons of more than two groups. *p* < 0.05 was considered significant.

## Figures and Tables

**Figure 1 ijms-24-08103-f001:**
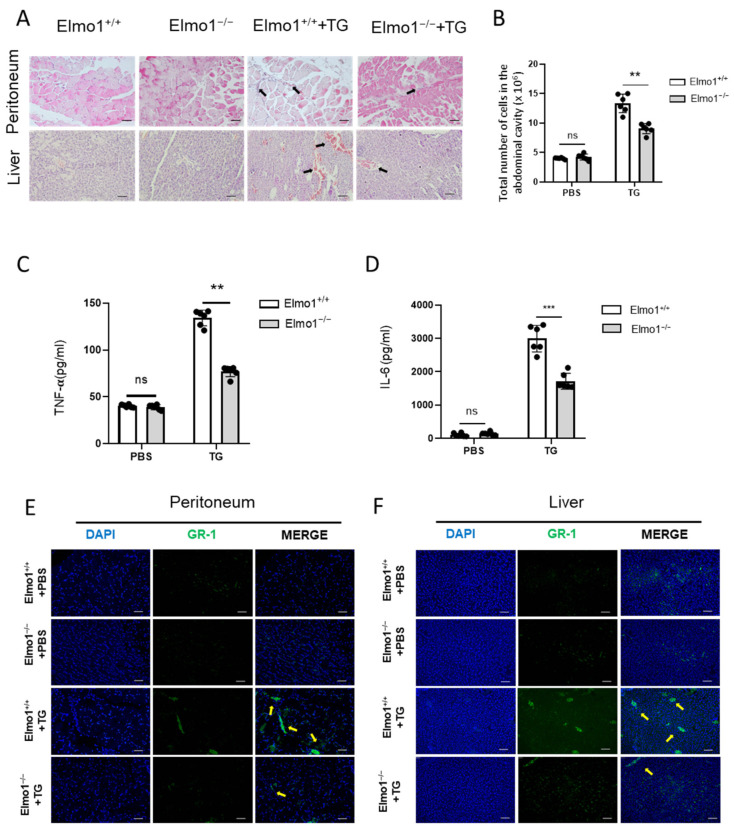
ELMO1 plays a role in acute peritonitis by regulating infiltration of neutrophils. (**A**) H&E-stained peritoneum and liver sections. The inflammatory cell infiltration is indicated by black arrows. Scale bar, 50 μm. (**B**) Total number of cells in peritoneal lavage fluid. The number of TG-induced cells in peritoneal lavage fluid were detected by flow cytometry. (**C**,**D**) Relative content levels of TNF-α and IL-6 in peritoneal lavage fluid detected by ELISA. The data are presented as means ± SEM (*n* = 6). (**E**,**F**) Immunofluorescence-stained histological peritoneum and liver sections. Yellow arrows indicate inflammatory cell infiltration. Scale bar: 50 μm. Statistical significance was assessed by *t*-test, ** *p* < 0.01, *** *p* < 0.001, and ^ns^
*p* > 0.05. *n* = 6.

**Figure 2 ijms-24-08103-f002:**
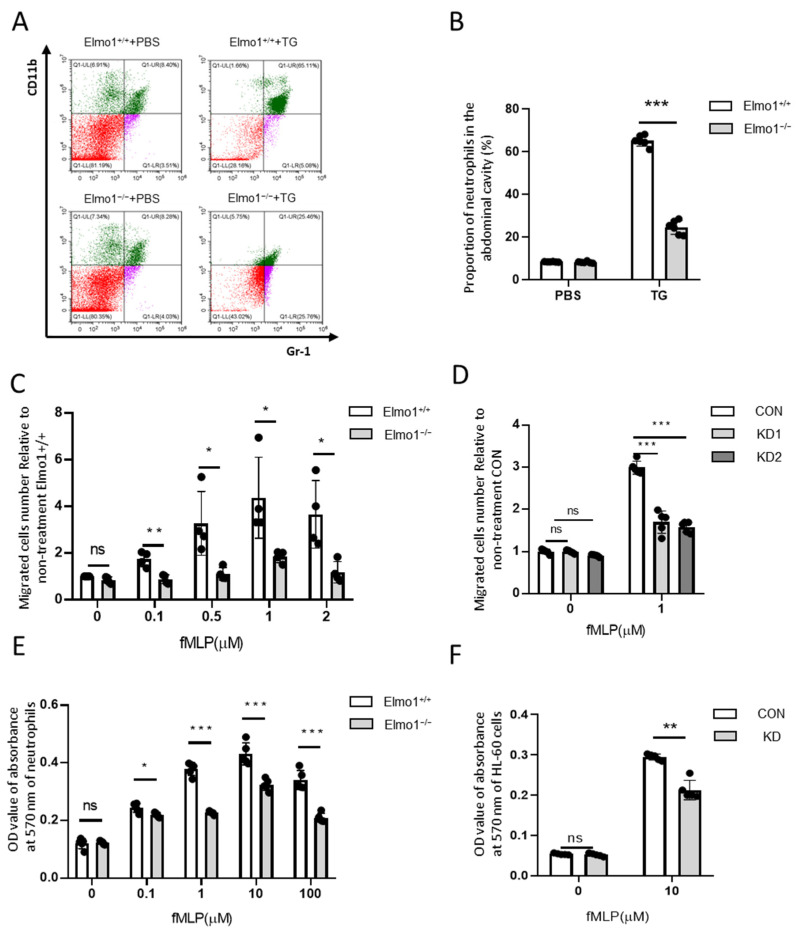
Elmo1 deficiency reduces the chemotaxis and adhesion of neutrophils. (**A**) The percentage of neutrophils number in the peritoneal lavage fluid. TG-induced neutrophil aggregation was stained with CD11b and Gr-1 to detect the ratio by flow cytometry (different colors means different cell population). (**B**) The percentage of CD11b- and Gr-1-positive cells were quantified with FlowJo software and the data are presented as means ± SEM (*n* = 5). (**C**,**D**) Primary neutrophils (*n* = 4) and HL-60 cells (*n* = 5) migration assay. Cells migrate under stimulation of fMLP for 2 h. Migrated cell numbers in the bottom chamber were counted by flow cytometry. The data are presented as means ± SEM. (**E**,**F**) Primary neutrophils and HL-60 cells adhesion assay. The fMLP-stimulated cells were incubated at 37 °C for 2 h and stained with crystal violet followed by 2% SDS dissolving for measuring absorbance at 570 nm. The data are presented as means ± SEM (*n* = 5). Each group had 3 replicates. Statistical significance was assessed by *t*-test, KD groups versus CON group by one-way ANOVA. * *p* < 0.05, ** *p* < 0.01, *** *p* < 0.001, and ^ns^
*p* > 0.05.

**Figure 3 ijms-24-08103-f003:**
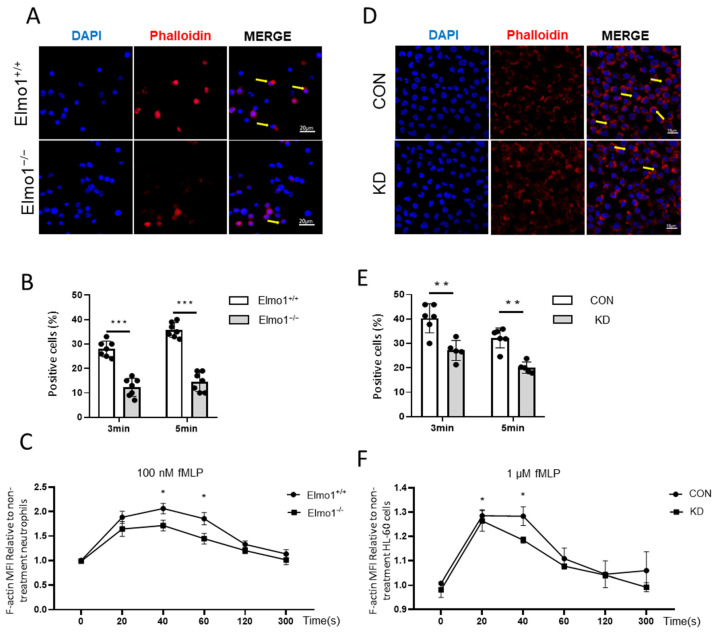
Deficiency of Elmo1 down-regulates fMLP-induced actin polymerization. (**A**,**B**) Neutrophil F-actin polymerization assay. Confocal micrographs of fMLP-stimulation cells stained with Alexa Fluor 633 phalloidin and DAPI. The percentage of phalloidin-positive cells in vision were quantified with ImageJ software and the data are presented as means ± SEM (*n* = 7). (**C**) The fMLP-stimulated primary neutrophils were stained with phalloidin to detect the F-actin by flow cytometry (*n* = 5). (**D**,**E**) F-actin polymerization assay in HL-60 cells (*n* = 5). (**F**) The fMLP-stimulated HL-60 cells F-actin polymerization assay (*n* = 5). MFI: mean fluorescence intensity. Yellow arrows indicate F-actin polymerization cells. Each group had 3 replicates. Statistical significance was assessed by *t*-test, * *p* < 0.05, ** *p* < 0.01 and *** *p* < 0.001.

**Figure 4 ijms-24-08103-f004:**
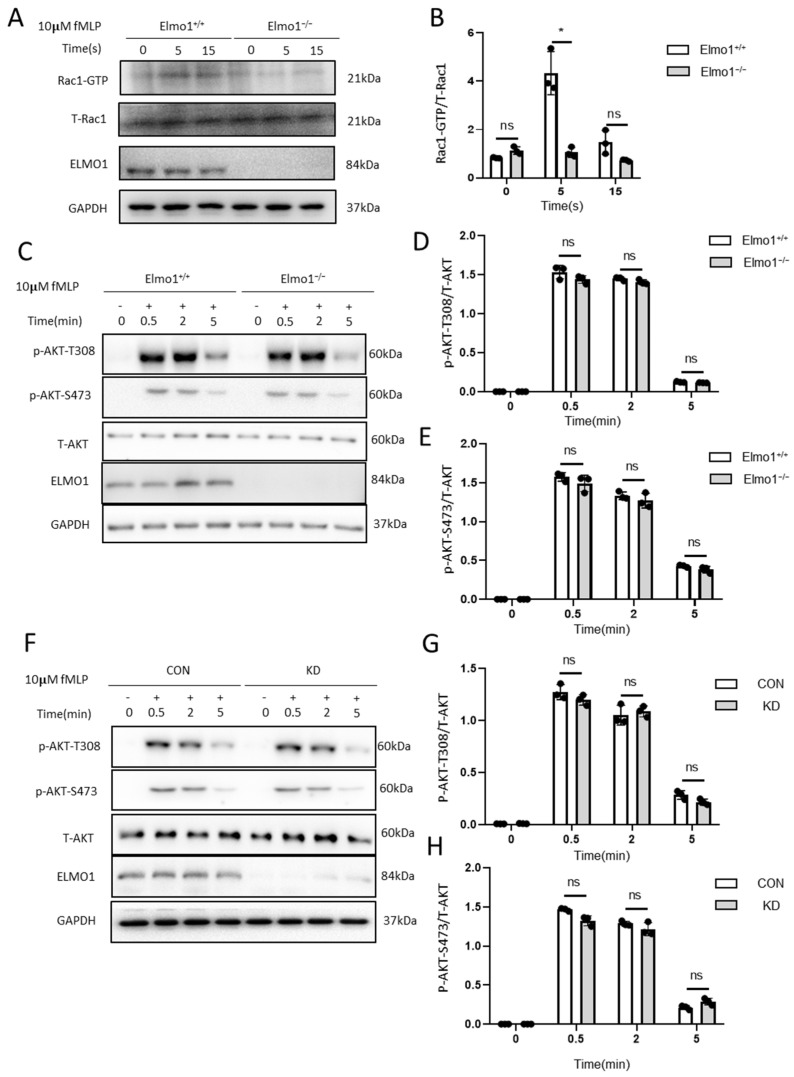
ELMO1 regulates fMLP-triggered Rac activation independent of PI3K and mTORC2 signaling pathways. (**A**) fMLP-induced Rac1 activation assay. Total proteins of neutrophils were collected after 5 and 15 s of fMLP stimulation, incubated with PAK-PBD beads and then subjected to SDS-PAGE and analyzed by immunoblotting for detecting for Rac1. (**B**) The ratio of Rac1-GTP to total Rac1 was quantified and graphed. Data are presented as mean ± SEM representation (*n* = 3). (**C**,**F**) fMLP-stimulated neutrophils and HL-60 cells. Phosphorylated AKT and total AKT were detected by SDS-PAGE Western blotting. (**D**–**H**) The levels of phosphorylated AKT-T308 (*n* = 3) and AKT-S473 (*n* = 3) were density-determined using ImageJ software and expressed as the ratio of phosphorylated form to total protein. Data are presented as means ± SEM. Each group had 3 replicates. Statistical significance was assessed by *t*-test, * *p* < 0.05, and ^ns^
*p* > 0.05.

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
