# Peer review of "ELMO1 Deficiency Reduces Neutrophil Chemotaxis in Murine Peritonitis"

_ijms, 2023, doi:10.3390/ijms24098103_

Round 1

Reviewer 1 Report

In this manuscript by Yu et al, the authors utilize Elmo1 knockout mice to examine how loss of Elmo1 effects the migration of neutrophils to the peritoneum, using thioglycolate (TG)-induced peritonitis. The authors first show that acute peritonitis is suppressed by loss of Elmo1, along with infiltration of neutrophils in the liver, each in response to TG.  Inflammatory signaling mediators TNFa and IL-6 are also reduced, as are the numbers of Gr-1 labeled neutrophils. Based on these data and the role Elmo1 might play in neutrophil chemotaxis, the authors use their Elmo KO mouse model (and derived primary neutrophils) plus HL-60 cells that overexpress shRNA directed against ELMO-1 to show that neutrophils exhibit deficient chemotaxis, as they are recruited to the peritoneal cavity or in response to fMLP. Finally, they show that f-actin polymerization is deficient in the Elmo1 KO neutrophils, as is Rac1-GTP activation, but the loss of Elmo1 does not appear to affect activation of PI3K or mTORC2 pathways (via analyses of phospho-AKT specific to each pathway). The data are convincing (despite issues with several figures and descriptions, see below), and certainly reveal a potential role for ELMO1 in human inflammation involving neutrophil migration, and therefore may be important to infections that follow peritoneal dialysis. However, there are some issues that should be addressed by the authors as follows.

Lines 46-52, the authors might include mention of the connection between the indicated signaling components to f-actin polymerization, as this was a major finding in their results showing loss of Elmo1 disrupts neutrophil chemotaxis.

Line 64, the authors should mention how adhesion fits into the signaling pathway that regulates chemotaxis, and its importance to chemotaxis.

Fig 1B and text (lines 74-75): how was flow cytometry used to assess total numbers from the peritoneal lavage? How were the ELISA assays for TNF-a and IL-6 performed? These methods were not described in the methods section and are required. 

Figs 1E and 1F, the images of DAPI-stained cells in E are too faint (those in 1F could also be more illuminated).

Fig 2C and D, how is flow cytometry used in the chemotaxis assay, and what are the dimensions of the numbers shown in the Y axis? And what is used to determine “relative” values, i.e., what are the values relative to? What are the values of the X-axis (assume fMLP, but this should be shown in the figure with uM concentrations).

Figures 2E and F, again what are the values in the Y-axes "relative" to? What are the units, if this is from absorbance readings? This needs to be described better in either the Results and/or Methods sections.

Lines 134-135, that authors should explain how staining with Phalloidin will reveal f-actin polymerization, and indicate this polymerization (e.g., add arrows in the images to indicate the positive cells).

Figures 3C and 3F, and in the text that describes these figures, what is being shown by the figure? What is the MFI and the shown values, what are the values for the Y axis, and how does this indicate differences in adhesion? This is not described in either the text or the figure legend. This should also be detailed in the Materials and Methods (this reviewer did not understand how this was performed from the methods included, other than the cells were stained with the fluorescence-tagged phalloidin).

Figure 4A, were these cell lysates really analyzed at 5 and 15 seconds after fMLP stimulation, or is it minutes? All other assays were shown in minutes. Also, do the authors have a loading control for this series of probings, e.g., GAPDH? This would be good to show.

Figure 4B, the X-axis should be labeled with Time and units, as should 4D, E, G and H.

Lines 240-242, the authors should mention the concentration of DMSO used to differentiate HL-60 cells. 

The authors might consider adding to the Discussion (and even the Abstract) a comment about how the identified role of ELMO1 in neutrophil chemotaxis might be utilized to design treatments that will inhibit their infiltration into the peritoneum, which may help with kidney failure patients.

 Overall: The manuscript requires a careful review for grammatical errors, as there are numerous issues throughout the manuscript. Below are just two examples.

Lines 16-18, the authors might rearrange the first sentence of the abstract since the subject of the ending phrase "which is characterized by an increase in neutrophil infiltration" is confusing as to which this refers to, the peritoneal inflammation or the process of performing peritoneal dialysis.

Line 39, the "in" in the phrase "to the site of infection in a four-step process including.....", should be "is".

Reviewer 2 Report

The role of ELMO1 in neutrophil chemotaxis has been recently described in inflammatory arthritis PMID: 30643265. Authors here confirm this observation in mouse model thioglycollate-induced peritonitis. In my opinion, this manuscript will be greatly benefited from the addition of data showing the effect of Elmo1 on disease onset and severity. My comments are below.

1.       In several experiments author has only analyzed 3 samples per group. This is a very small sample size. 3 is the bare minimum to get any statistics. Authors should add a minimum of 6 samples per group and show dot plots of every sample.

2.       Author should mention how many independent experiments were performed per experiment. Figure 1E. DAPI images needs more exposure.

3.       Figure 2- Apart from fMLP author should also test other chemotactic agents such as leukotriene B4 (LTB4), platelet activating factor, and complement-derived C5a to test if the defect in chemotaxis is not limited to specific agent.

4.       Figure 3A- Please show single and merge channels separately. As many of these cells do not have typical doughnut shaped neutrophil nuclei, author should stain these samples for Ly6G or MPO to indicate neutrophils in the sample. What was the purity after neutrophil isolation?

5.       What is the impact of Elmo1 deletion on disease activity? The author should investigate the impact of Elmo1 deletion on disease onset and severity.

6.       Line 246 “After stimulated, the cells were fixed……. “Please correct the grammar. 

Reviewer 3 Report

The manuscript addresses one of the key aspects of neutrophil activity, their ability to recruit in response to chemotactic factors, which determines the course of response to the presence of infectious agents and inflammation. This is an important issue given the role of neutrophils in peritonitis. Unfortunately, the planned research profile has some limitations.

First, the introduction needs serious additions, for example:

- what is the mechanism of chemotaxis in general, role of Rho and Rac

- what factors regulating chemotaxis, particularly in neutrophils

- known chemoattractants for neutrophils

Secondly, testing the proteins of only one PI3K signaling pathway seems insufficient and greatly diminishes the value of the study. 

Fig.3. Incomplete description of the figure C and F.

A short and uninteresting discussion summarizing the results. The importance of the observations made, eg in various states of pathology, should be pointed out.

Reviewer 4 Report

The manuscript is well-written and interesting. However, major issues should be considered for improvement:

- English editing (necessary),

- size of each protein bands should be displayed (none of them has it!),

- please adapt the blots so we can follow the protein bands ordinary,

- please describe the methods properly and mention the Catalogue Number, Mansufacturer and Country of the chemicals, antibodies, etc.; this may be of critical importance since it leads to the reproducibility of you data by other groups;

- the Discussion is insufficient since many aspects are ad-hoc discussed without proper elaboration (you should extend the discussion of your findings, show whether other published works have indiciated similar findings and elaborate any putative clinincal appications of these findings),

- one interesting aspect is that fMLP triggers also neutrophil activation as reported by many reports which may lead to the activation of PI3K/Akt. This pathway is also regulated by TLR4 by a "pathogen"-dependent route. The authors should discuss any interplay between these pathways which may be of fundamental importance for their findings;

- another interesting aspect of neutrophils remaine the induction of memory-like features which may be implicated in various inflammatory responses (also during the PD). Authors should consider to discuss this aspect! Here you may find the following papers as very helpful (please don't stick precisely to these papers, you may cite other relevant studies):

1) Review (here they also expressed updated information in regard to PD and memory-like functions): https://doi.org/10.3390/biomedicines11030766

2) Role of gram-positive LPS promoting neutrophil memory reactions affecting transmigratory and phagocytic activities: https://doi.org/10.3390/biomedicines11030766

3) Role of gram-negative LTA promoting adaptive manners affecting migratory properties: Gram-positive Staphylococcus aureus LTA promotes distinct memory-like effects in murine bone marrow neutrophils - ScienceDirect

4) The role of microbiome promoting adaptive manners in murine neutrophilic cells: Biomedicines | Free Full-Text | Gut Microbiota-Derived Small Extracellular Vesicles Endorse Memory-like Inflammatory Responses in Murine Neutrophils (mdpi.com)

- also a very important issue is to included more then 3 Experiments (it is not sufficient to provide data only from n=3), please consider to increase the number of experiments since they may lead to a realistic effect. As a researcher, sometimes n=3 may be a "lucky" experience!

- please specify why did you use the Student T-test for your setup? Usually for this kind of setup I would expect the usage of 1-Way-ANOVA as proper statistical test!

Round 2

Reviewer 1 Report

The authors have addressed the concerns of this reviewer and have improved the quality and impact of the manuscript.

Reviewer 2 Report

Accept in current form. 

Reviewer 4 Report

The authors improved the Ms substantially. Thanks!